# Beyond Identity: High-Fidelity Face Swapping by Preserving Source Video Attributes

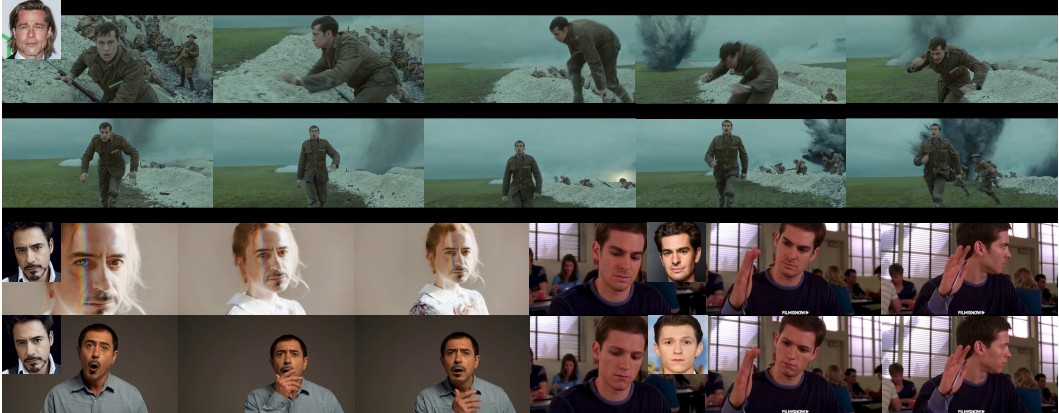

Figure 1: Qualitative results of our proposed video reference guided face swapping model, LivingFace. Our method achieves stable identity preservation across long video sequences, while faithfully inheriting source-video attributes such as lighting, expressions, and viewpoints. Compared to existing approaches, LivingFace demonstrates strong robustness under challenging conditions and generalizes well across diverse identities and source videos.

## Abstract

Video face swapping is crucial in film and entertainment production, where achieving high fidelity and temporal consistency over long and complex video sequences remains a significant challenge. Inspired by recent advances in reference-guided image editing, we explore whether rich visual attributes from source videos can be similarly leveraged to enhance both fidelity and temporal coherence in video face swapping. This work presents LivingFace, the first video reference guided face swapping model. Our approach employs keyframes as conditioning signals to inject the target identity, enabling flexible and controllable editing. By combining keyframe conditioning with video reference guidance, the model performs temporal stitching to ensure stable identity preservation and high-fidelity reconstruction across long video sequences. To address the scarcity of data for reference-guided training, we construct a paired face-swapping dataset, Face2FaceSwap, where the generated data are fed as inputs and the original data serve as ground truth, thereby enabling reliable supervision. Extensive experiments demonstrate that our method achieves state-of-the-art results, seamlessly integrating the target identity with the source video's expressions, lighting, and motion, while significantly reducing manual effort in production workflows.

## 1 Introduction

Video face swapping holds significant value in the film and entertainment industries. However, existing methods fall short of meeting the stringent demands of high-quality cinematic production. For instance, GAN-based approaches (Li et al., 2019; DeepFakes, 2020; Chen et al., 2020; Shiohara et al., 2023; Luo et al., 2025), which typically process videos frame-by-frame (see Fig. 2), have made

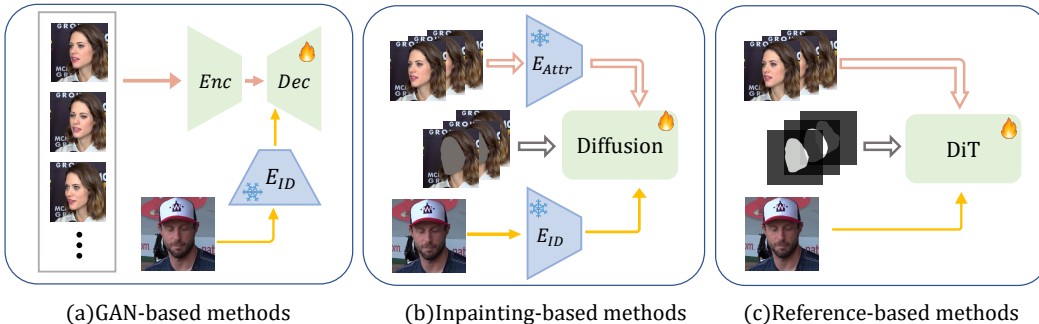

(a)GAN-based methods     (b)Inpainting-based methods     (c)Reference-based methods

Figure 2: (a) GAN-based approaches process videos in a frame-by-frame manner, and therefore often suffer from temporal inconsistency. (b) Inpainting-based methods focus on generating the facial region based on sparse conditions, which inevitably leads to a loss of fidelity and unnatural visual artifacts. (c) Recent reference-based generation methods enable faithful utilization of rich visual attributes contained in references and demonstrate remarkable capability in preserving them.

notable progress in injecting target identity but often suffer from temporal inconsistencies—such as flickering and jitter—especially in long sequences. Meanwhile, contemporary video diffusion models (Zhao et al., 2023; Han et al., 2024; Chen et al., 2024; Wang et al., 2025), while achieving high visual quality and temporal consistency, often rely on sparse conditioning signals such as facial landmarks. This reliance makes it challenging to perfectly align the generated expressions, lighting, and subtle nuances with the source video, resulting in faces that may appear unnatural or lack lifelike vitality. Consequently, there is a critical need for a video face swapping model capable of directly leveraging the rich, detailed information from the source video's facial region.

Achieving a high degree of customization while preserving the integrity of the original content remains a fundamental challenge in generative media (Yang et al., 2023a). Methods based on DDIM inversion (Ju et al., 2023; Qi et al., 2023; Geyer et al., 2023) or Score Distillation Sampling (SDS) (Poole et al., 2022; Hertz et al., 2023) often struggle to strike an optimal balance between editability and fidelity. In the field of video editing, a common strategy involves combining inpainting with structural guidance such as depth or keypoints (Jiang et al., 2025; Hu et al., 2025; Tu et al., 2025). However, such approaches inherently discard the original pixel information within the edited region, leading to a noticeable loss of fidelity in details.

Recently, reference guided generation has demonstrated remarkable breakthroughs in image editing, successfully reconciling editing flexibility with high-fidelity reconstruction (Labs et al., 2025; Deng et al., 2025; Wu et al., 2025). This approach directly guides the model using the reference images, enabling the faithful utilization of rich visual attributes contained in the references. Nevertheless, adapting these techniques to video face swapping presents unique challenges: (1) the scarcity of paired training data for reference-guided video face swapping task; and (2) the difficulty of injecting a stable and consistent identity condition throughout long and complex video sequences.

In this work, we address these challenges by introducing LivingFace, the first video editing model for face swapping that directly references the source video's details. To facilitate this, we construct Face2FaceSwap, the first-of-its-kind dataset specifically curated for video reference face swapping. Meanwhile, we reverses the data pairs to ensure reliable ground-truth supervision. Furthermore, we decompose the challenging task of long-video face swapping into a highly controllable pipeline comprising keyframe selection, identity injection, video completion, and temporal stitching. Extensive experiments demonstrate that our approach achieves state-of-the-art results, seamlessly blending the target identity with the high-definition details of the source video, including its original expressions and lighting conditions.

Our contributions are multi-faceted. We provide a detailed analysis of the impact of the generated data distribution on face swapping performance, and demonstrate the critical role of data diversity in determining model effectiveness. Benefiting from its highly controllable pipeline and superior generation quality, LivingFace is uniquely suited for the demands of the professional film and television industry. It can incorporate meticulous manual editing results while drastically reducing the intensive labor costs associated with frame-by-frame processing.

## 2 RELATED WORK

**Video face swapping.** The task of video face swapping is to replace the identity in a video while preserving attributes such as pose, expression, illumination, and background. GAN-based approaches (Li et al., 2019; DeepFakes, 2020; Chen et al., 2020; Shiohara et al., 2023; Luo et al., 2025), which typically process videos frame-by-frame, have made notable progress in injecting target identity through encoder–decoder pipelines, feature matching, or two-stage refinement. However, they often suffer from temporal inconsistencies—such as flickering and jitter—especially in long sequences. Recently, video diffusion models are used in video face swapping. Diffusion-based methods (Zhao et al., 2023; Han et al., 2024; Chen et al., 2024; Wang et al., 2025) demonstrate stronger generative power and achieve higher visual quality and temporal consistency. They treat face swapping as inpainting by masking the original face and regenerating it with a diffusion model conditioned on background frames and auxiliary attribute encoding (Chen et al., 2024). This often leads to the loss of fine-grained details and introduces inconsistencies with the model's pretrained priors, thereby degrading generation quality. In this work, we tackle these challenges by directly leveraging detailed source video references for face swapping, combined with a carefully curated dataset Face2FaceSwap and an inverse data training strategy to provide high-fidelity supervision.

**Diffusion-based Video Editing.** With the rapid progress of diffusion models, a variety of video editing methods have been developed that can be broadly categorized into inversion-based, inpainting-based, and reference-guided approaches. Inversion-based methods (Ju et al., 2023; Qi et al., 2023; Geyer et al., 2023; Poole et al., 2022; Hertz et al., 2023) reconstruct the original video trajectory in the diffusion process to enable editing, but they often struggle to balance editability and fidelity. Inpainting-based approaches (Jiang et al., 2025; Hu et al., 2025; Tu et al., 2025) edit masked regions with structural guidance such as optical flow, depth, or keypoints, achieving temporal coherence but usually at the cost of losing fine-grained details. Recently, reference-guided methods (Labs et al., 2025; Deng et al., 2025; Wu et al., 2025; Hurst et al., 2024; Comanici et al., 2025) have shown strong potential by leveraging reference images or frames to combine flexible editing with high-fidelity reconstruction. Nonetheless, extending this paradigm to long video sequences remains challenging due to the scarcity of paired data and the difficulty of maintaining consistent identity or attributes over time. In this work, we use reference-guided generation for face video swapping, enabling controllable identity transfer while preserving temporal coherence and visual fidelity across long sequences.

## 3 PRELIMINARY: VIDEO GENERATION WITH DiT AND RECTIFIED FLOW

Recent advancements in diffusion-based video generation leverage the Diffusion Transformer (DiT) architecture combined with continuous-time training objectives such as Rectified Flow (RF) to achieve high-quality and temporally coherent synthesis. DiT extends traditional UNet-based diffusion backbones with transformer blocks, enabling more flexible and scalable modeling of high-dimensional video data. In this framework, the model learns a continuous denoising process by predicting the velocity between a pair of latent points. Given a ground-truth sample $x_1$, and a standard Gaussian noise $x_0 \sim \mathcal{N}(0, I)$, a linearly interpolated latent $x_t$ is constructed as:

$$x_t = tx_1 + (1 - t)x_0, \tag{1}$$

where $t \in [0, 1]$ is a timestep sampled from a predefined distribution. The target velocity is defined as the derivative of $x_t$ with respect to time, yielding:

$$v_t = \frac{dx_t}{dt} = x_1 - x_0. \tag{2}$$

The DiT model is trained to estimate this velocity given the latent $x_t$, the conditioning signal $c$, and the timestep $t$. Let $u(x_t, c, t; \theta)$ be the model's predicted velocity, where $\theta$ denotes the model parameters. The training objective is to minimize the mean squared error (MSE) between the predicted and ground-truth velocities:

$$\mathcal{L} = \mathbb{E}_{x_0, x_1, c, t} \left\| u(x_t, c, t; \theta) - v_t \right\|^2. \tag{3}$$

This training formulation enables high-quality results with significantly fewer steps and greater computational efficiency in video generation.

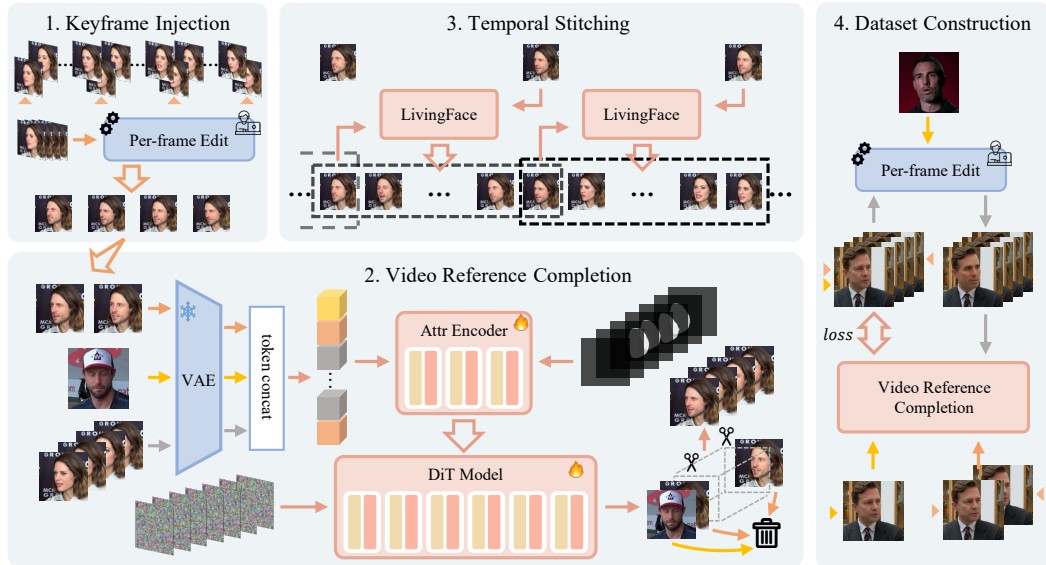

Figure 3: Overview of the proposed LivingFace framework for video face swapping. (1) High-quality keyframes are selected under favorable conditions and used as temporal anchors to ensure consistent identity injection across long sequences. (2) We directly feed the full-pixel source video as a reference, enabling high-fidelity reconstruction of non-identity attributes such as lighting and micro-expressions. (3) By sequentially generating chunks and propagating the final frame of the previous chunk as guidance, LivingFace achieves seamless transitions in long videos. (4) We use per-frame edit method to generate the data and invert data roles to construct paired samples, ensuring reliable and artifact-free learning.

## 4 METHOD

In video face swapping tasks, the input typically consists of a source video $V_s = \{f_t \mid t \in [1, T]\}$ to be modified, a mask sequence $M = \{m_t \mid t \in [1, T]\}$ indicating the target regions for editing, and a target identity image $I_{\text{tar}}$. The overall designs of LivingFace are illustrated in Fig. 3. In the following sections, we introduce the designs of LivingFace focusing on four fundamental components of video face swapping: target identity injection, source video attribute preservation, consistent long-video generation, and paired-dataset construction.

### 4.1 KEYFRAMES IDENTITY INJECTION

Effectively injecting target identity into the swapped results is the basic challenge in face swapping tasks. Previous face swapping methods typically encode the target identity into a identity vector using a pre-trained identity encoder (Wang et al., 2018; Deng et al., 2019a), which is then injected into the model. The model is trained to decode this identity vector into the edited facial region, achieving satisfactory identity similarity with the target face. However, in long and complex video sequences, this approach often suffers from identity flickering across frames due to motion variations, resulting in temporal inconsistency.

To address the aforementioned issues, we draw inspiration from frame interpolation paradigms and combine them with strengths of image-based face swapping in industrial settings, designing a keyframe-based identity injection scheme. Guided by discrete keyframes within a video, diffusion-based methods using frame interpolation paradigms typically generate temporally smooth and stable long sequences (Wang et al., 2024). Meanwhile, image-based face swapping has become a reliable solution in production environments. By applying per-frame editing — typically involving face-swapping models combined with post-processing tools such as PhotoShop — high-quality results can be achieved under favorable conditions. These conditions include frontal views, simple lighting, and neutral facial expressions. Building on both, we design a method that achieves stable

identity injection via keyframe selection and guidance. First, we select video frames meeting industrial face-swapping criteria as keyframes to obtain high-quality single-frame swapping inputs $F_{\text{key}}^{\text{swap-in}} = \{f_{k_i}^{\text{swap-in}} \mid k_i \in \mathcal{K}, \mathcal{K} \subset [1, T]\}$. Then, as illustrated in Fig. 3, we take a pair of adjacent keyframes and use them as the boundary frames to guide the generation of the intermediate face-swapped sequence, including the two reference keyframes. Additionally, a target face image $I_{tar}$ is used to fill possible identity gaps in the first or last keyframe (e.g. occlusion or closed eyes). This identity injection strategy effectively leverages the temporal priors of video generation models to maintain identity consistency throughout the sequence. It also yields robust results on challenging intermediate frames where image-based face-swapping methods often fail.

## 4.2 Video Reference Completion

In addition to identity injection, it is crucial to incorporate both non-identity attributes of the edited region and the unaltered content of the source video into the model. Previous diffusion-based video face-swapping methods that follow an inpainting formulation typically reconstruct the attributes of the edited region using structural features extracted from the source video (Han et al., 2024; Wang et al., 2025). However, this approach discards the original pixel information within the edited region, resulting in a noticeable loss of fine-grained fidelity. Moreover, as diffusion models are not pretrained to interpret such structural features, the required additional training often compromises their generative priors.

Inspired by the success of reference-guided methods in image editing, we extend this paradigm to video face swapping for high-fidelity reconstruction. As illustrated in Fig. 3, instead of masking the facial region in the source video and relying on external pretrained encoders, we directly input the correspond full-pixel source video chunk $V_s^{[k_i:k_{i+1}]} = \{f_t \mid t \in [k_i, k_{i+1}]\}$ as the visual reference. This design allows the model to preserve the detailed visual attributes, such as lighting and micro-expressions, without degradation. Together with identity signals, we encode each conditional input using the VAE encoder $\mathcal{E}_\phi(\cdot)$, and concatenate the resulting latent token sequences along the token dimension in the following order:

$$Z_c = \text{Concat}_{\text{token}}\big(\mathcal{E}_\phi(I_{\text{tar}}), \mathcal{E}_\phi(f_{k_i}^{\text{swap-in}}), \mathcal{E}_\phi(V_s^{[k_i:k_{i+1}]}), \mathcal{E}_\phi(f_{k_{i+1}}^{\text{swap-in}})\big), \tag{4}$$

where $Z_c$ denotes the aggregated conditional representation in latent space. This ordered concatenation aligns well with the temporal modeling of video diffusion models, allowing the generative process to be guided by priors across time. To further support spatial localization, we construct the binary mask sequence $M$ with black-filled masks to indicate the regions to be edited, and concatenate it with $Z_c$ along the channel dimension.

For adaptive feature injection, we introduce an attribute encoder composed of DiT blocks that share the same architecture as the diffusion backbone. These blocks are initialized with the corresponding pretrained weights. The output of each attribute encoder layer is added element-wise to the corresponding layer of the backbone, enabling latent-space conditioning in a layer-wise manner. Formally, the injection process is defined as:

$$X^{(l+1)} = \mathcal{D}_\theta^{(l)}\Big(X^{(l)} + \mathcal{A}_\psi^{(h)}(Z_c^{(h)}, M)\Big). \tag{5}$$

where $X^{(l)}$ denotes the hidden representation at layer $l$ of the DiT backbone $\mathcal{D}_\theta$, and $\mathcal{A}_\psi^{(h)}$ is the $h$-th block of the attribute encoder with parameters $\psi$. This formulation enables adaptive conditioning while preserving the pretrained generative priors of the diffusion model.

## 4.3 Temporal Stitching

To meet the needs of industrial face swapping for videos of variable length, we split long videos into multiple fixed-length chunks. Generating each chunk independently often causes noticeable jumps between adjacent chunks. Fortunately, thanks to our keyframe design and the incorporation of video reference guidance, we achieve smooth transitions using a temporal stitching method. Specifically, we process chunks sequentially in temporal order. When generating a middle chunk, we use the last frame output $f_{k_i}^{\text{swap-out}}$ of the previous chunk instead of $f_{k_i}^{\text{swap-in}}$ as the first-frame guidance, while the last-frame guidance continues to use $f_{k_{i+1}}^{\text{swap-in}}$:

$$\{f_t^{\text{swap-out}}\}_{t=k_i}^{k_{i+1}} = \mathcal{D}_{\theta,\psi}\big(f_{k_i}^{\text{swap-out}}, f_{k_{i+1}}^{\text{swap-in}}, V_s^{[k_i:k_{i+1}]}, I_{\text{tar}}, M\big), \tag{6}$$

| Methods | ID Retri. ↑ | ID Simil. ↑ | Expr.↓ | Lighting↓ | Gaze↑ | Pose↓ | FVD↓ | Avg. Rank↓ |
|---|---|---|---|---|---|---|---|---|
| Deepfakes | 0.893 | 0.432 | 2.941 | 0.340 | 0.584 | 4.662 | 47.54 | 9.28 |
| FaceShifter | 0.918 | 0.485 | 2.451 | 0.225 | 0.690 | 2.696 | 18.73 | 5.00 |
| InfoSwap | 0.961 | 0.542 | 2.868 | 0.290 | 0.586 | 2.962 | 47.28 | 7.14 |
| SimSwap | **0.989** | 0.562 | 2.674 | 0.221 | **0.720** | 2.977 | 33.97 | 4.57 |
| BlendFace | 0.929 | 0.480 | 2.256 | 0.228 | 0.717 | 2.196 | 21.96 | 4.28 |
| CanonSwap | 0.946 | 0.523 | 2.307 | 0.205 | 0.685 | **1.782** | 30.30 | 4.00 |
| DiffSwap | 0.435 | 0.261 | **1.912** | **0.199** | 0.687 | 2.277 | 83.98 | 5.71 |
| Face-Adapter | 0.501 | 0.247 | 2.564 | 0.259 | 0.641 | 3.608 | 36.83 | 8.28 |
| inswapper | 0.968 | **0.636** | 2.536 | 0.214 | 0.704 | 2.464 | 20.63 | 3.71 |
| LivingFace (Ours) | 0.973 | 0.592 | 2.466 | 0.211 | 0.706 | 2.336 | 19.29 | **3.00** |

Table 1: Quantitative comparison with state-of-the-art methods on FF++. For each metric, the top-3 methods are highlighted in cyan, while the others are shown in gray. Lower values indicate better performance for ↓ metrics (Expr., Lighting, Pose, FVD), and higher values indicate better performance for ↑ metrics (ID Retrieval, ID Similarity, Gaze). The last column reports the average ranking across all metrics, with our method achieving the best overall performance.

For the first chunk, both the first and last frame guides are given by the respective keyframe inputs. Moreover, to allow flexible selection of keyframe positions under the constraint of fixed-chunk inference length, we also employ techniques such as frame interpolation, reverse playback, and frame skipping combined with multiple inference as needed.

## 4.4 DATASET CONSTRUCTION

Face video datasets typically contain only single videos of individuals and lack paired source–target samples for face swapping. Consequently, these datasets cannot be directly used to train our method as reference-guided methods relies on paired source–target samples to supervise the extraction of non-identity attributes from the source video. To obtain such pairs, we generate synthetic data using existing face-swapping models. Compared with diffusion-based methods, GAN-based face swapping approaches achieve better fidelity to the source video by leveraging the full-pixel source video as input. However, GAN-based results frequently exhibit temporal inconsistencies and degraded visual quality (e.g., artifacts and distortions). If such issues dominate the synthetic data, the resulting supervision would be unreliable, leading to ineffective training of our method. To overcome this challenge, as illustrated in Fig. 3, we invert the data roles: the GAN-generated swapped video serve as the model input $V_s$, while the original video provides the keyframe inputs $F_{\text{key}}^{\text{swap-in}}$, the target image $I_{tar}$ and the ground truth supervision. This design guarantees that the reference and the ground truth share the same identity, while also providing artifact-free, high-quality supervision signals.

## 5 EXPERIMENTS

### 5.1 EXPERIMENTAL SETUP

**Dataset.** For training, we construct our dataset Face2FaceSwap based on CelebV-Text (Yu et al., 2023) and VFHQ (Xie et al., 2022). CelebV-Text is a large-scale video–text dataset containing approximately 70,000 in-the-wild facial video clips, totaling around 279 hours of footage. VFHQ (Video Face High-Quality) comprises over 16,000 high-resolution video clips collected from YouTube, covering diverse scenarios and identities, with frame sizes typically ranging from 700×700 to 1000×1000. Based on these two datasets, we synthesize paired face-swapping data to build our training dataset, Face2FaceSwap. For evaluation, we adopt FaceForensics++ (FF++) (Rossler et al., 2019), a widely used benchmark for face manipulation analysis. FF++ consists of 1,000 pristine video sequences and includes variations in compression levels and resolutions, providing a challenging and realistic testbed for assessing both fidelity and robustness.

**Metrics.** To comprehensively evaluate the face-swapping performance, we employ both image-level and video-level metrics to assess the quality of the generated results. Following prior work Chen et al. (2020; 2024); Wang et al. (2025), we randomly sample 10 frames from each face-swapped

| Methods | ID Retri. ↑ | ID Simil. ↑ | Expr.↓ | Lighting↓ | Gaze↑ | Pose↓ |
|---|---|---|---|---|---|---|
| LivingFace | **0.948** | **0.536** | 2.84 | 0.285 | 0.451 | 2.84 |
| VACE | 0.572 | 0.313 | 3.08 | 0.355 | 0.299 | 6.42 |
| w/o Target Image | 0.927 | 0.515 | 2.74 | 0.279 | **0.537** | **2.80** |
| w/o Keyframe | 0.559 | 0.281 | **2.47** | **0.249** | 0.502 | 2.84 |
| Inpainting | 0.941 | 0.519 | 2.89 | 0.292 | 0.491 | 2.87 |

Table 2: Ablation of key components. Replacing video reference with inpainting reduces fidelity, highlighting its role in preserving non-identity attributes. Removing keyframe guidance decreases identity similarity and temporal consistency, while omitting target image reference also lowers identity accuracy under challenging conditions.

| Methods | ID Retri. ↑ | ID Simil. ↑ | Expr.↓ | Lighting↓ | Gaze↑ | Pose↓ |
|---|---|---|---|---|---|---|
| LivingFace | **0.948** | 0.536 | 2.84 | **0.285** | 0.451 | **2.84** |
| VACE | 0.572 | 0.313 | 3.08 | 0.355 | 0.299 | 6.42 |
| Using Upper Data | 0.943 | 0.532 | **2.82** | 0.289 | 0.484 | 2.89 |
| Using Lower Data | 0.947 | **0.540** | 2.83 | 0.288 | **0.488** | 2.87 |

Table 3: Ablation on generated data quality. Results show that identity performance remains stable across groups, but using the full dataset achieves better fidelity (e.g., gaze, pose) due to greater sample diversity, which enhances the model's robustness to identity variations.

video to compute image-level evaluation metrics, including ID Similarity, ID Retrieval, Expression Error, Lighting Error, Gaze Error, and Face Pose Error. ID Similarity is measured by encoding both the face-swapped result and the target image into identity vectors using a pre-trained ID encoder (Wang et al., 2018), followed by computing the cosine similarity between them. For ID Retrieval, the identity vector of the swapped image is compared against all samples in the dataset using cosine similarity to determine whether the correct target identity can be successfully retrieved. In addition to identity-related metrics, we calculate Expression and Lighting Errors by extracting their respective coefficients using a 3DMM-based face reconstruction method (Deng et al., 2019b) and computing the L2 distance between the source and swapped results. Similarly, we use a gaze estimation model (Abdelrahman et al., 2023) and a head pose estimation model (Ruiz et al., 2018) to predict gaze direction and head pose in the original and swapped frames, and compute the L2 distance to quantify the changes. For video-level evaluation, we use Frechet Video Distance (FVD) (Unterthiner et al., 2018) to assess the overall quality of generated videos, and Warping Error to evaluate motion consistency between the original and swapped sequences.

**Implementation Details.** As a recently released open-source video editing model, VACE (Jiang et al., 2025) builds upon the Wan video generation framework and is fine-tuned into a high-quality framework capable of controllable, customizable, and inpainting-based video editing. By loading its pre-trained weights and fine-tuning the model on our constructed dataset, we adapt VACE to the reference-guided video face-swapping task. Specifically, we train the model for 10,000 steps using the AdamW optimizer, with a learning rate of 1e-5 and a batch size of 16. The input resolution is set to 640, consistent with the preprocessing applied during dataset construction, and the number of frames is set to 81, following the original VACE configuration. All training experiments are conducted on 8 NVIDIA H200 GPUs. During inference, we first detect faces using a face detection model, then crop the detected regions and perform face swapping on the cropped sequences. The swapped regions are subsequently pasted back into their original positions within the frames. As detailed in Sec. 4.3, each video sample is divided into multiple chunks, which are processed sequentially. Following our training setup, each chunk contains 81 frames, and the cropped regions are resized to 640×640 before face swapping. As a face-swapping model commonly used in industrial scenarios, we employ Inswapper (Henry, 2025) for processing keyframes.

## 5.2 ABLATION STUDIES

**Ablation of Synthetic Data Quality.** As discussed in Sec. 4.4, while invert the data role ensures a matched identity and high-quality ground truth (GT), the generated data used as training input still

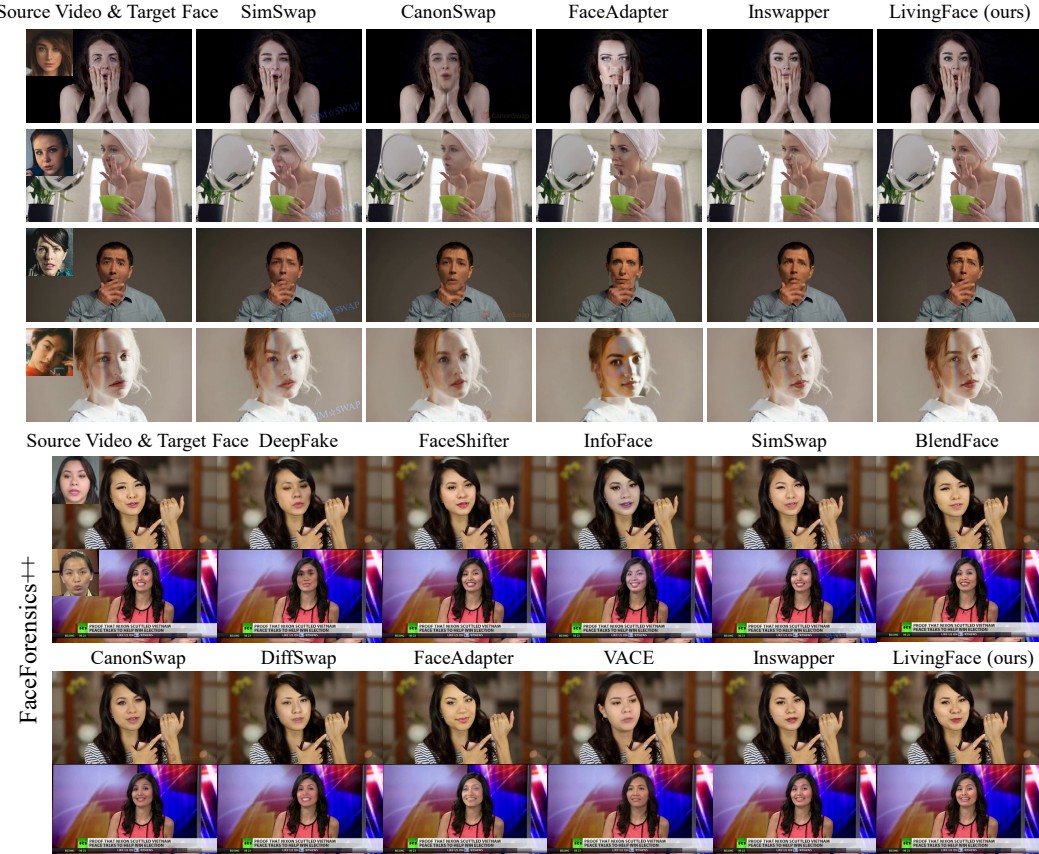

Figure 4: Quantitative and qualitative comparison with state-of-the-art face-swapping methods. LivingFace achieves the best overall performance across metrics and average ranks, surpassing both GAN- and diffusion-based approaches in video consistency, fidelity, and identity similarity. Despite using InSwapper for keyframe generation, our model delivers more stable results and better preserves source attributes, even under challenging conditions such as side profiles and occlusions.

suffers from the quality problem. On one hand, when the face-swapping model fails, the results often exhibit high identity similarity with the original video and may suffer from flickering. On the other hand, when there is a substantial discrepancy in identity, the resulting face swap exhibits artifacts. To investigate the impact of synthetic data quality on the final model performance, we conduct an ablation study. Since the aforementioned issues are closely linked to identity variation between the original video and synthetic data, we evaluated their ID cosine similarity. By visualizing the data distribution and analyzing samples from different intervals, we confirm that this metric effectively captures the two aforementioned scenarios (see Fig. 5). Consequently, we filter the data pairs based on this metric and categorize them into three groups for comparison: using the entire dataset, the first 70% of the data, and the last 70% of the data. The experimental results, presented in Table 3, show that neither the ID-similar data nor the data with significant ID differences have a substantial impact on the model's identity performance. However, we observe that using the full dataset yields stronger performance in terms of gaze, pose, and other fidelity metrics. We hypothesize that the inclusion of more diverse samples in the full dataset contributes to this outcome. This diversity enables the model to adapt to a broader range of identity-related variations, thereby enhancing its ability to preserve source attributes more robustly. Therefore, in our training process of next experiments, we use all available data pairs for training.

**Ablation of Model Design.** We conduct ablation studies on three key components of our model design: video reference, keyframe guidance, and target image reference. As shown in Table 2, when we replace the video reference with the traditional inpainting approach, the model exhibits a notable decline in fidelity metrics, including gaze, pose, lighting, and expression. This demonstrates that

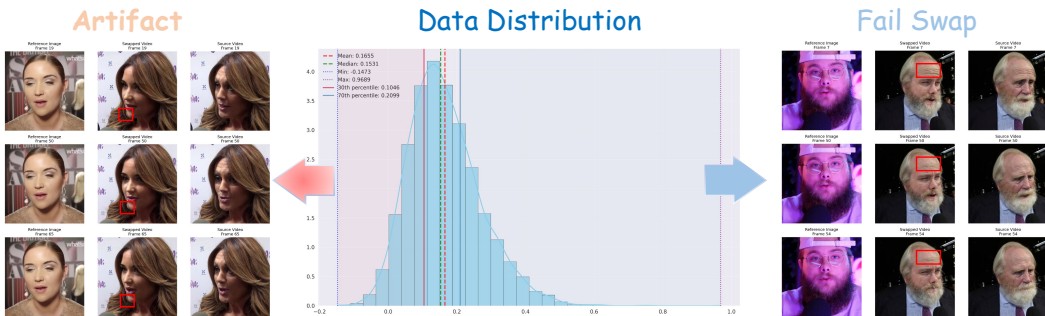

Figure 5: Visualization of the **Face2FaceSwap** dataset. The central plot shows the distribution of identity similarity scores, with the lowest 30% (red) and highest 30% (blue) highlighted. Low-similarity pairs often contain artifacts (left), while high-similarity pairs may retain reference information and reduce fidelity (right).

the video reference approach more effectively utilizes detailed information from the original video compared to the structural condition of inpainting, thereby enabling better preservation of non-identity attributes. Regarding identity injection, when we remove the keyframe guidance and rely solely on the target image, we observe a significant drop in identity similarity as well as a noticeable degradation in temporal consistency. This validates the effectiveness of keyframe guidance in ensuring stable identity injection across long video sequences. Conversely, when we ablate the identity information provided by the target image, we still observe a decline in identity similarity. This is due to the limitations of keyframes in certain scenarios—such as occlusion, extreme angles, or closed eyes—which may result in the loss of critical identity features.

## 5.3 COMPARISONS WITH EXISTING METHODS

In this section, we compare several state-of-the-art face-swapping methods, including Sim-Swap (Chen et al., 2020), InfoSwap (Gao et al., 2021), BlendSwap (Shiohara et al., 2023), CanonSwap (Luo et al., 2025), DiffSwap (Zhao et al., 2023), FaceAdapter (Han et al., 2024), as well as our baseline model, VACE (Jiang et al., 2025), and the widely used industrial face-swapping model, InSwapper (Henry, 2025), which is also employed for generating keyframes for our model. As shown in Table 1, LivingFace achieves state-of-the-art performance across multiple metrics and average ranks. Compared to our keyframe generation model, InSwapper, although the keyframes are generated based on its outputs, our model demonstrates superior video consistency, better preservation of source video attributes, and more stable face-swapping results in challenging scenarios such as side profiles and occlusions (as shown in Fig. 4). This also indicate our model exhibits strong robustness when handling problematic keyframes, as detailed in Supplementary Material. Additionally, GAN-based methods such as SimSwap and CanonFace exhibit poor performance in video quality and consistency, while diffusion-based methods do not perform well in terms of fidelity and identity similarity. These conclusions from the quantitative experiments align with our qualitative results, as shown in Fig. 4.

## 6 CONCLUSION

This work presented LivingFace, the first video reference-guided face swapping model that leverages keyframes as conditioning signals to enhance both fidelity and temporal coherence in video face swapping. By combining keyframe conditioning with video reference guidance, our approach ensures stable identity preservation and high-fidelity reconstruction across long video sequences. We propose a novel paired dataset, Face2FaceSwap, and an inverse training strategy, offering reliable ground-truth supervision, tackling the challenge of scarce data for reference-guided training. Extensive experiments validate that our method sets a new state-of-the-art, seamlessly integrating target identities with source video expressions, lighting, and motion. Our model significantly reduces manual effort in production workflows, enabling more efficient and flexible video editing in film and entertainment.

## 7 ETHICS STATEMENT

This work adheres to the ICLR Code of Ethics. In this study, no human subjects or animal experimentation was involved. All datasets used, including Face2FaceSwap, VoxCeleb, FaceForensics++ and CelebV-HQ, were sourced in compliance with relevant usage guidelines, ensuring no violation of privacy. We have taken care to avoid any biases or discriminatory outcomes in our research process. No personally identifiable information was used, and no experiments were conducted that could raise privacy or security concerns. Since video face swapping has potential for misuse (e.g., deepfakes), we restrict the use of our dataset and code strictly to academic research and prohibit malicious applications such as disinformation, harassment, or unauthorized impersonation. We are committed to maintaining transparency, integrity, and responsible AI practices throughout the research process.

## 8 REPRODUCIBILITY STATEMENT

We have made every effort to ensure that the results presented in this paper are reproducible. All code, pre-trained models, and the Face2FaceSwap dataset will be made publicly available upon acceptance to facilitate replication and verification. The experimental setup, including training steps, model configurations, optimizer details, hyperparameters, and hardware specifications, is described in detail in the paper. We have also provided a full description of our inverse training strategy, the keyframe conditioning module, and temporal stitching pipeline to assist others in reproducing our experiments. Additionally, publicly available datasets such as VoxCeleb, FaceForensics++, and CelebV-HQ are used for training and evaluation, ensuring consistent and comparable results across studies. We believe these measures will enable other researchers to reproduce our work faithfully and further advance the field of controllable video face swapping.

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

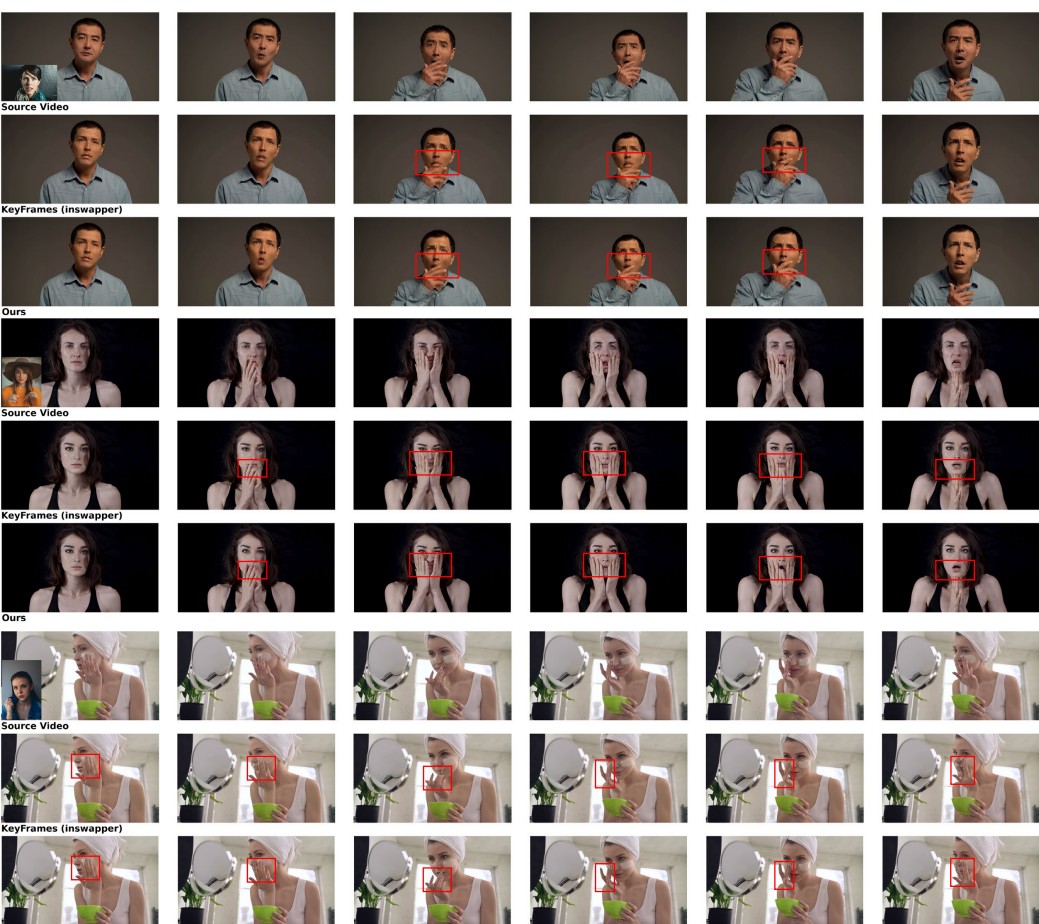

Figure 6: Qualitative comparison under challenging scenarios including occlusions, exaggerated expressions, and large shape deformations. While the keyframe-based face-swapping baseline (*InSwapper*) often produces artifacts or distorted facial regions (red boxes), our method successfully refines the keyframes and achieves temporally consistent and high-fidelity results across diverse conditions.

## A   USE OF LLMS

We used large language models (LLMs) only for minor assistance in polishing the language and adjusting the presentation of tables. No LLMs were involved in designing the methodology, conducting experiments, or analyzing results.

## B   ROBUSTNESS IN KEYFRAME QUALITY

As shown in Fig. 6, our model demonstrates strong robustness in terms of keyframe quality. Even when our keyframe face-swapping model, InSwapper, produces suboptimal results in challenging scenarios such as occlusions or side profiles, we are able to refine its outputs by regenerating the keyframes. Furthermore, when using other less effective models as the keyframe preprocessing network, as shown in Table 1, our model consistently outperforms the corresponding models, achieving better overall results.

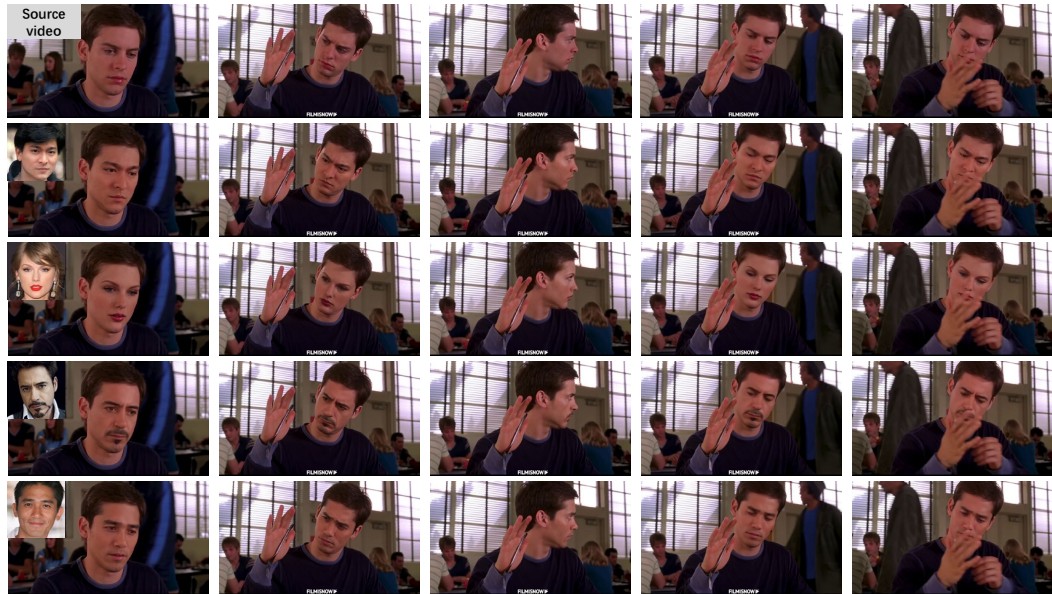

Figure 7: Identity swapping results on the same source video with different target identities. Our method produces consistent and high-fidelity face swaps regardless of large or small identity differences, demonstrating strong robustness to identity variations.

## C  ROBUSTNESS TO IDENTITY DIFFERENCES

For the scenario of swapping different identities for the same source video, we conducted experiments with multiple videos and identities. As shown in Fig. 7, leveraging the advantages of image-based identity injection, LivingFace achieves satisfactory results for the same video, regardless of whether the identity difference is large or small. We hypothesize that this robust of identity difference is due to the diversity of identities in our training data, as discussed in Sec. 5.2.

## D  ROBUSTNESS TO ATTRIBUTE VARIATIONS IN SOURCE VIDEO

To verify whether our reference-based video face swapping approach is robust to attribute variations in the source video, we selected a diverse set of videos as source inputs and conducted experiments using the same target identity. As shown in Fig. 8, our model consistently produces high-quality results across attributes in challenging scenarios, such as occlusions, side profiles, and complex lighting conditions. Furthermore, owing to the robustness of keyframe quality, our model is able to generate realistic, high-fidelity outputs even when the keyframe model produces suboptimal results.

## E  COMPARISON WITH CLOSE-SOURCE METHODS

Recently, several inpainting-based video face swapping methods using the Stable Video Diffusion model (Blattmann et al., 2023) are proposed, such as HiFiVFS (Chen et al., 2024) and FaceAdapter (Han et al., 2024). However, these methods are not open-source. To enable a comparison with them, we captured several demos from their project websites and conducted tests using the same target face image. The comparative results are shown in the Fig. 9. Our approach better preserves the original video attributes such as lighting and expression, and also demonstrates strong stability in occluded cases.

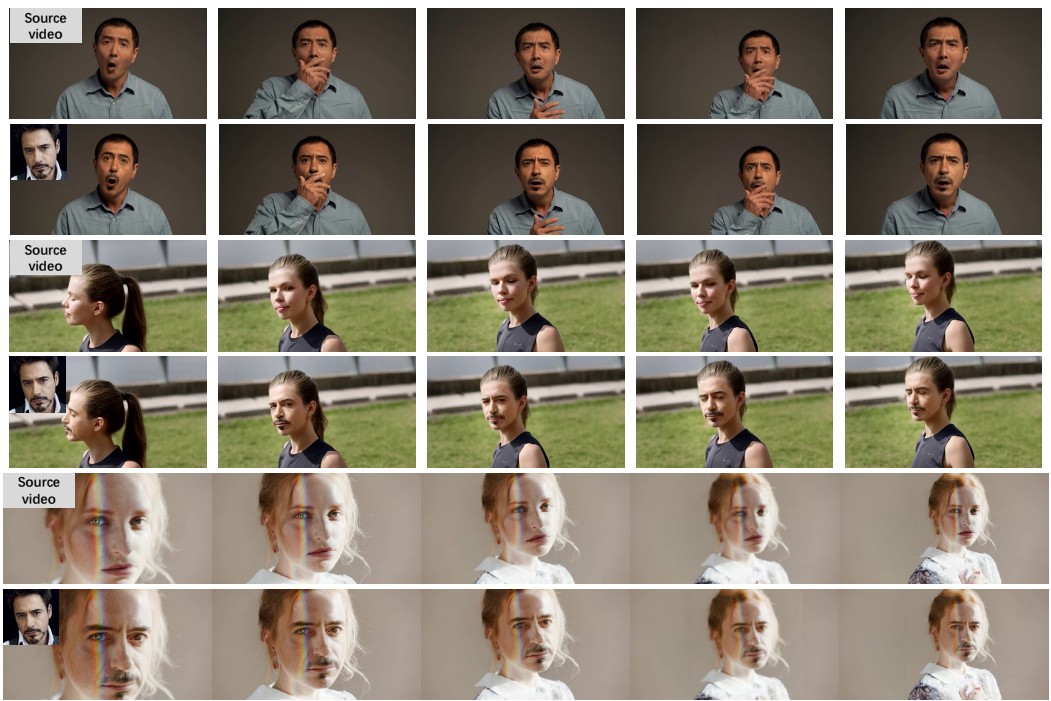

Figure 8: Face swapping results on diverse source videos with the same target identity. Our method consistently preserves target identity and produces high-fidelity outputs across challenging conditions, including occlusions, side profiles, and complex lighting.

## F  FACE2FACESWAP CONSTRUCTION DETAILS

We construct our dataset Face2FaceSwap based on CelebV-Text (Yu et al., 2023) and VFHQ (Xie et al., 2022). First, we perform crop, resize, and clipping operations on the dataset to ensure the resolution is 640×640 pixels and the video length is approximately 200 frames. We then randomly pair the data and extract the first frame from the target video as the target face image. Next, we apply InSwapper (Henry, 2025) to perform face-swapping on the entire dataset. The process is conducted using 8 NVIDIA H100 GPUs over a duration of 120 hours. Additionally, we use the face-parsing model (Yu et al., 2018) to generate the face mask video. For the ablation study on the inpainting paradigm, we also use the pose estimation model (Yang et al., 2023b) to generate the corresponding pose video. After filtering out the failed samples from the preprocessing steps, our dataset Face2FaceSwap contains a total of 152,221 video samples, with a cumulative duration exceeding 300 hours.

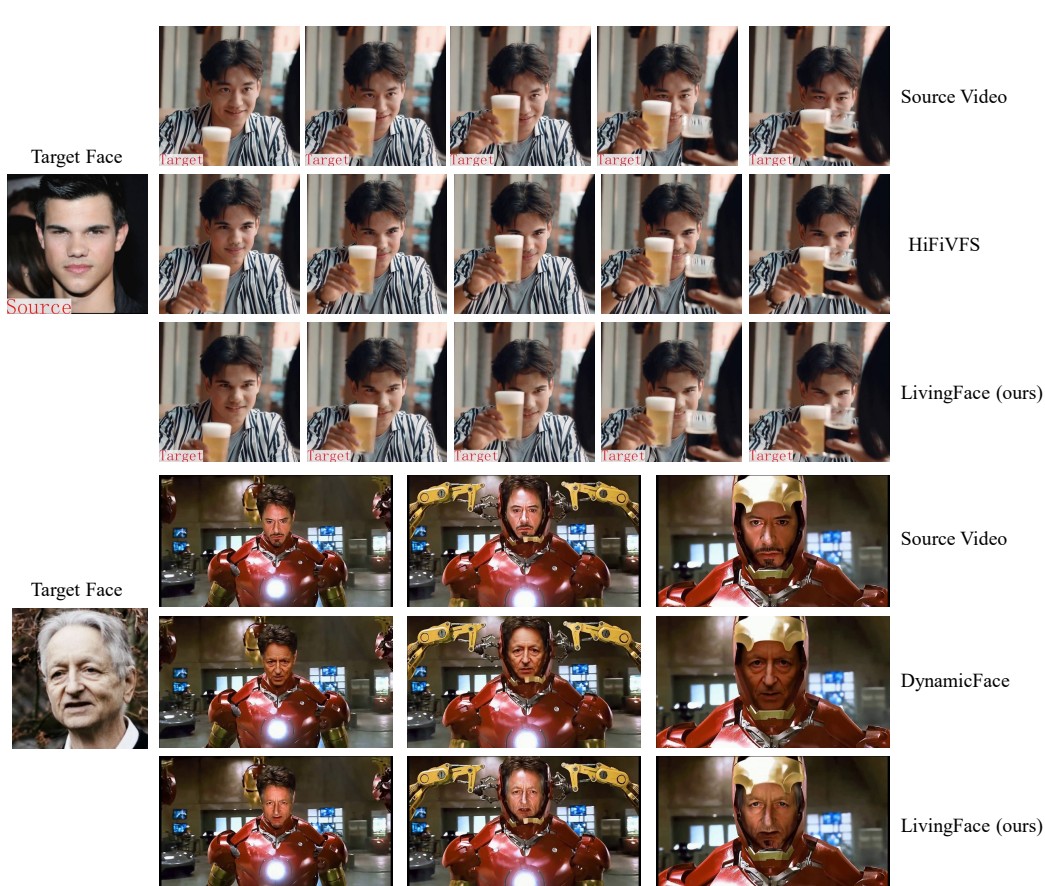

Figure 9: Qualitative results.

