# OpenReview forum: "Beyond Identity: High-Fidelity Face Swapping by Preserving Source Video Attributes"
_ICLR.cc/2026/Conference — ICLR 2026 Conference Withdrawn Submission_

### Official Review · Reviewer_ivSL · 2025-10-21

**Soundness:** 3
**Presentation:** 3
**Contribution:** 3
**Rating:** 6
**Confidence:** 3

**Summary:**

This paper proposes LivingFace, aiming to address the core challenges of low fidelity and poor temporal consistency in long video sequences for film and entertainment production. The model leverages keyframes as conditioning signals to inject target identities and combines them with full-pixel source video references to preserve non-identity attributes. To solve the scarcity of paired training data for reference-guided tasks, the authors construct the Face2FaceSwap dataset using an inverse data strategy—treating GAN-generated swapped videos as inputs and original videos as ground truth. Additionally, the model uses temporal stitching to ensure smooth transitions in long videos split into fixed-length chunks. Extensive experiments on the FaceForensics++ (FF++) benchmark show that LivingFace outperforms state-of-the-art methods (e.g., InSwapper, SimSwap) across metrics like ID Retrieval, Expression Error, and FVD, achieving both stable identity preservation and high-fidelity attribute retention.

**Strengths:**

- Framework Design: LivingFace fills the gap in reference-guided video face swapping, beating frame-by-frame GAN methods (prone to temporal inconsistency) and sparse-condition diffusion methods (prone to detail loss). By directly using full-pixel source videos as references, it effectively preserves fine-grained attributes like micro-expressions and lighting.
- Dataset: The Face2FaceSwap dataset and inverse training strategy are practical and effective. By reversing the roles of GAN-generated swapped videos (inputs) and original videos (ground truth), the model obtains high-quality, artifact-free supervision—addressing a critical bottleneck for reference-guided training.
- Applicability: The keyframe-based identity injection and temporal stitching modules are tailored to real-world needs. They support variable-length long videos, reduce manual frame-by-frame editing effort, and maintain robustness in challenging scenarios (occlusions, side profiles), making it suitable for professional film/TV production.

**Weaknesses:**

- The model relies on InSwapper (an industrial face-swapping tool) to generate high-quality keyframes. While the paper mentions robustness to suboptimal keyframes, it does not clarify how the model performs if the keyframe generator (e.g., InSwapper) fails severely (e.g., extreme occlusions, blurred faces), which limits its adaptability to low-quality source videos.
- Training requires 8 NVIDIA H200 GPUs, and inference splits videos into 81-frame chunks. The paper lacks analysis of computational efficiency (e.g., inference time per frame, memory usage) compared to lightweight methods (e.g., InSwapper, SimSwap), which may hinder deployment in resource-constrained scenarios.
- The model focuses on preserving facial attributes (lighting, expressions) but does not discuss how it handles non-facial context (e.g., hair, accessories, background shadows linked to facial lighting). Mismatches in these contexts could reduce overall realism.

**Questions:**

- How does LivingFace perform when the keyframe generator (e.g., replacing InSwapper with a lower-performance model like DeepFakes) produces low-quality keyframes?
- Can the model be optimized for efficiency without sacrificing performance? What is the trade-off between speed and quality in practical deployment?

---

### Official Review · Reviewer_7MER · 2025-10-29

**Soundness:** 2
**Presentation:** 1
**Contribution:** 2
**Rating:** 2
**Confidence:** 4

**Summary:**

This paper, "Beyond Identity: High-Fidelity Face Swapping by Preserving Source Video Attributes" presents a method for face swapping in videos using a pre-trained Diffusion Transformer model. The method uses manually chosen keyframes to inject idenity into the face swapping, followed by the video reference completion and video stitching. A contribution is to separate the elements to edit as identity based and non-identity based. The visual results seem good compared to the previous work.

**Strengths:**

The visual results are good, and show a clear improvement with respect to the state of the art. The basis of the model is VACE, itself based on a powerful state of the art video generation model (Wan), so this of course helps the method.

**Weaknesses:**

While the writing of the paper initially seemed quite good, when I tried to understand in greater detail, I found it to be vague and confusing. More generally, the presentation of the method requires a lot of work in my opinion, and is difficult to follow in the details. I would find it extremely difficult to implement the paper.

**Questions:**

- Section 4.1: In this Section, you select keyframes to inject identity. In general, this section is very vague, I did not find any point where you actually describe how the identity is injected. In Figure 3 it is just shown as "Per-frame edit", but never said how this is done. It is quite frustrating to read. Furthermore you say "we seleect video frames meeting industrial face-swapping criteria". What are these criteria ? How are the frames chosen ? If the process is not automatic, then it makes the process significantly heavier, so this should be really clear.

- Section 4.4, the dataset construction. In this section, you say that there is not enough data with source/target pairs of the same person, and that you therefore (line 297) "generate synthetic data using face-swapping models". This seems extremely problematic ! You are basing your method on a database which is created with another editing method, so how can you hope to do better than this method ? While I accept that this is just to extract the non-identity attributes, it is still problematic. Furthermore, to compound the problem, you do not even say which method you used to do this (or I could not find a reference) ! And finally, you do not compare to this method, which would be the minimum to do. Or if you did compare to the method, we cannot know, since you never give the name of the method. Even if there is a reason why the first argument (you cannot do better than your dataset) is not valid, I would have at least expected you to discuss it.

- In Figure 3, you have an arrow saying "loss" but I can find nowhere in the paper where you explain this loss ? Is this just the reconstruction loss ? You should at least say this once. The inly time you talk about training is in section 5, the experiments, in the "Implementation Details" paragraph. It seems to me that stating that the model is fine-tuned and how it is fine tuned is more than a simple implementation detail.

- Section 5.3, line 465: "As shown in Table 1, Living face achieves state-of-the-art performance across multiple metrics and average ranks". This is not true !! In none of the metrics are you better than the state-of-the-art. I do not at all think that it is necessary to beat the state of the art consistently in every situation, nor even in any situation if your method possesses some useful attribute (faster, simpler to implement/understand etc.).

Other details:

- p.3, line 161. "significantly fewer steps": fewer than what ? (diffusion I guess, but it is vague).
- p.5, line 238 : "directly input the correspond full-pixel" -> "directly input the corresponding full-pixel"

---

### Official Review · Reviewer_uQpR · 2025-11-01

**Soundness:** 1
**Presentation:** 2
**Contribution:** 2
**Rating:** 2
**Confidence:** 3

**Summary:**

This manuscript presents a video face-swapping model that uses reference-swapped keyframes. Specifically, the authors first utilize image face-swapping models to edit keyframes in the source video. After this, the edited keyframes, the full source video chunk, and the identity image latents are concatenated, and their features are extracted with a trainable copy of the main DiT model (as an attribute encoder). The extracted features are then injected into the main DiT model to generate the face-swapped videos. Meanwhile, the authors also utilize a per-frame editing method to construct the training data. From the experimental results, the proposed framework achieves competitive performance compared to existing face-swapping models.

**Strengths:**

This manuscript considers face attribute preservation, which is often neglected by previous masked inpainting methods. It would be very helpful if the authors could open-source the constructed dataset.

**Weaknesses:**

1. Although the authors claim they try to better preserve the attributes of the source video by conditioning on the full source video, the reviewer is not convinced this can be achieved by the proposed framework. The authors constructed the "inverse" dataset using existing per-frame swapping methods, using the swapped video as input. However, with this strategy, the swapped video (which serves as the input) and the original video (which serves as the GT) cannot be guaranteed to have the same attributes. Therefore, using these data pairs for training does not ensure the preservation of attributes from the source video. Moreover, the input videos (the GAN-swapped results) are not temporally consistent, which will further enlarge the gap between the training and inference stages.

2. The proposed pipeline is not an end-to-end solution. It has a critical dependency on a pre-existing per-frame face swapper (e.g., Inswapper) to generate the initial keyframes. If the keyframe editing fails, the final swapped video will likely fail as well.

3. Regarding the evaluation results, the quantitative metrics do not show clear superiority over existing methods. Meanwhile, the reviewer recommends that the authors also report the model's runtime to better demonstrate its practical advantages.

4. Regarding the ablation study, the authors should also report results using different per-frame face-swapping models to edit the keyframes. Furthermore, the method for selecting which keyframes to edit is not clearly explained in the manuscript.

**Questions:**

See the weakness.

---

### Official Review · Reviewer_ooc1 · 2025-11-03

**Soundness:** 3
**Presentation:** 3
**Contribution:** 2
**Rating:** 4
**Confidence:** 4

**Summary:**

This paper introduces LivingFace, a video face swapping framework leveraging video reference guidance and keyframe-based identity injection to enable high-fidelity, temporally consistent face replacement in long video sequences. The approach concatenates full-source video frames and key identity frames—augmented with a novel attribute encoder—for controllable, high-quality synthesis in challenging cinematic scenarios. To support supervised training, the work proposes a new paired dataset (Face2FaceSwap) constructed via an inverse training strategy. Extensive experiments show state-of-the-art performance across multiple quantitative metrics and qualitative comparisons with prior art.

**Strengths:**

1.Superior Quantitative and Qualitative Results: Table 1 presents a comprehensive benchmark on FF++, with LivingFace ranking best on average across identity, expression, lighting, pose, and fidelity metrics. Major improvements are also illustrated by large-scale qualitative comparisons in Figure 4 and robustness analysis in Figure 6, indicating strong stability in occlusions, non-frontal poses, and challenging lighting.
2.Dataset Contribution: Introduction and detailed documentation of Face2FaceSwap, with a data construction methodology grounded in sound quality-control logic.

**Weaknesses:**

1.Limited Discussion of Generalization and Overfitting in Dataset Construction: Section 4.4 and Appendix F discuss constructing Face2FaceSwap by inverting GAN-based swaps. However, the paper provides no analysis of possible overfitting to synthetic data, or whether the method is robust to genuine out-of-distribution content.
2.While FVD and identity metrics are included, there is no perceptual study or user evaluation to assess results for film or entertainment audiences.
3.Section 4.2 introduces the DiT-attribute encoder fusion for adaptive conditioning. While the mechanism is intuitive, there is no detailed justification or analysis on the effectiveness or stability of this injection scheme. How are multi-scale representations aligned? Are there training instabilities or mode collapse issues when fusing attribute encodings at each layer?

**Questions:**

1.Can the authors provide qualitative/quantitative results on edge cases with rapid motion, heavy occlusion throughout, or drastic lighting changes? Are there systematic limitations observed in such cases?
2.How does the method perform when reference videos are of lower resolution, contain heavy occlusions, or do not cover the full pose/expression range present in the target video?
3.The stitching process between chunks is handled by passing forward the last generated frame, but no explicit discussion is given of error accumulation. Could this lead to identity drift or propagation of subtle artifacts in very long sequences?

---

### Note · Authors · 2025-11-14

I have read and agree with the venue's withdrawal policy on behalf of myself and my co-authors.